# Diffeomorphic Mesh Deformation via Efficient Optimal Transport for Cortical Surface Reconstruction

**Tung Le**[1], **Khai Nguyen**[2], **Shanlin Sun**[1], **Kun Han**[1], **Nhat Ho**[2] **& Xiaohui Xie**[1]
[1]University of California, Irvine
[2]The University of Texas at Austin

## Abstract

Mesh deformation plays a pivotal role in many 3D vision tasks including dynamic simulations, rendering, and reconstruction. However, defining an efficient discrepancy between predicted and target meshes remains an open problem. A prevalent approach in current deep learning is the set-based approach which measures the discrepancy between two surfaces by comparing two randomly sampled point-clouds from the two meshes with Chamfer pseudo-distance. Nevertheless, the set-based approach still has limitations such as lacking a theoretical guarantee for choosing the number of points in sampled point-clouds, and the pseudo-metricity and the quadratic complexity of the Chamfer divergence. To address these issues, we propose a novel metric for learning mesh deformation. The metric is defined by sliced Wasserstein distance on meshes represented as probability measures that generalize the set-based approach. By leveraging probability measure space, we gain flexibility in encoding meshes using diverse forms of probability measures, such as continuous, empirical, and discrete measures via *varifold* representation. After having encoded probability measures, we can compare meshes by using the sliced Wasserstein distance which is an effective optimal transport distance with linear computational complexity and can provide a fast statistical rate for approximating the surface of meshes. To the end, we employ a neural ordinary differential equation (ODE) to deform the input surface into the target shape by modeling the trajectories of the points on the surface. Our experiments on cortical surface reconstruction demonstrate that our approach surpasses other competing methods in multiple datasets and metrics.

## 1 Introduction

Mesh deformation is a fundamental task in 3D computer vision and computer graphics. A wide range of shape reconstruction tasks (Chen & Zhang, 2019; Jiang et al., 2020; Mescheder et al., 2019; Niemeyer et al., 2020; Park et al., 2019; Sun et al., 2023; Yariv et al., 2020) and shape registration (Ashburner, 2007; Dalca et al., 2018; Balakrishnan et al., 2019; Le et al., 2024; Krebs et al., 2019; Dalca et al., 2019; Han et al., 2024; Sun et al., 2022; Han et al., 2023) leverages state-of-the-art mesh deformation methodology. One popular approach for mesh deformation is to estimate the vertex displacement vectors (3D offsets) while keeping their connectivity (Wang et al., 2019; Bongratz et al., 2022). However, displacement-based methods cannot guarantee the manifoldness of the resulting mesh and often produce self-intersecting faces. To address this issue, diffeomorphic transformation (Ruelle & Sullivan, 1975; Arsigny, 2004) is one effective way to deform a mesh while preserving its topology. As an instance of diffeomorphic surface deformation, Neural Mesh Flow (NMF) (Gupta, 2020) learns a sequence of diffeomorphic flows between two meshes and models the trajectories of the mesh vertices as ordinary differential equations (ODEs). However, these methods have limited shape representation capacity and struggle to perform well on complex manifolds. To overcome this limitation, several diffeomorphic mesh deformation models have been proposed, such as CortexODE (Ma et al., 2022) and CorticalFlow (Lebrat et al., 2022), which aim to handle hard manifolds, such as the cortical surface. While CorticalFlow (Lebrat et al., 2022) introduces diffeomorphic mesh deformation (DMD) modules to learn a series of stationary velocity fields, CortexODE (Ma et al., 2022) encodes spatial information of the MRI images along

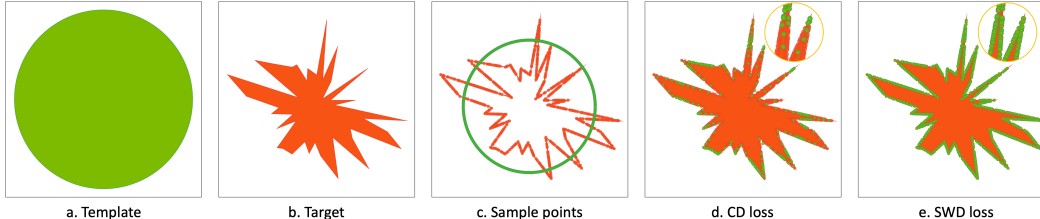

Figure 1: **2D deformation toy example**. We deform a (a) template circle to a (b) target polygon via an optimization-based setting. (c) Points are uniformly sampled from the two contours. CD loss is easily trapped at local minima, i.e. the green points are not uniformly distributed on the target contour as desired. In contrast, (e) SWD loss can find the optimal transport plan among discrete probability measures, i.e. the resulting points are distributed more uniformly along the contour. More details about this toy example can be found in the Appendix B.

with vertices features using an MLP model and employs neural ODEs (Chen et al., 2018) to model the trajectories of the points. Nonetheless, these approaches often rely on Chamfer distance as the objective function, which might have disadvantages.

Selecting an appropriate metric to evaluate the dissimilarity between two meshes is a crucial step in learning deformation mesh models. Recent literature favors the approach of using set-based comparison due to its simplicity. In particular, the set-based approach first samples two sets of points on the surface meshes, and then use Chamfer distance (CD) (Deng et al., 2018; Duan et al., 2019; Groueix et al., 2018) to compare two meshes and optimize them. However, the CD loss tends to get trapped in local minima easily (Achlioptas et al., 2018; Nguyen et al., 2021b; 2023; 2024a), failing to distinguish bad samples from the true ones, as demonstrated by our toy example in Fig. 1. Although a weighted CD has been proposed to prioritize fitting local regions with high curvatures in Vox2Cortex (Bongratz et al., 2022), the issue is only alleviated but not resolved completely, still resulting in suboptimal assignments between two sets of points. Therefore, we propose novel approaches to transform a mesh into a probability measure that generalizes the set-based approach. Furthermore, by relying on the probability measure approach, we can employ geometric measure theory to represent mesh as an oriented varifold (Almgren, 1966; Vaillant & Glaunes, 2005; Glaunès et al., 2008) and get a better approximation of the mesh compared to the random sampling approach. After that, we adopt efficient optimal transport to compare these measures since optimal transport distances are naturally fitted to compare disjoint-support measures.

Wasserstein distance (Peyré & Cuturi, 2019; Villani, 2009) has been widely recognized as an effective optimal transport metric to compare two probability measures, especially when their supports are disjointed. Despite having a lot of appealing properties, the Wasserstein distance has high computational complexity. In particular, when dealing with discrete probability measures that have at most $m$ supports, the time and memory complexities of the Wasserstein distance are $\mathcal{O}(m^3 \log m)$ and $\mathcal{O}(m^2)$, respectively. The issue becomes more problematic when the Wasserstein distance is computed on different pairs of measures as in mesh applications, namely, each mesh can be treated as a probability measure.

To improve the computational complexities of the Wasserstein distance, by adding entropic regularization and using the Sinkhorn algorithm (Cuturi, 2013), an $\epsilon$-approximation of Wasserstein distance can be obtained in $\mathcal{O}(m^2/\epsilon^2)$. However, this approach cannot reduce the memory complexity of $\mathcal{O}(m^2)$ due to the storage of the cost matrix. Moreover, the entropic regularization approach cannot lead to a valid metric between probability measures since the resulting discrepancy does not satisfy the triangle inequality. A more efficient approach based on the closed-form solution of Wasserstein is sliced Wasserstein distance (SWD) (Bonneel et al., 2015), which is computed as the expectation of the Wasserstein distance between random one-dimensional push-forward measures from two original measures. SWD can be solved in $\mathcal{O}(m \log m)$ time complexity while having a linear memory complexity $\mathcal{O}(m)$.

In this paper, we propose a learning-based **D**iffeomorphic mesh **D**eformation framework via an efficient **O**ptimal **T**ransport metric, dubbed **DDOT**, that learns continuous dynamics to smoothly deform an initial mesh towards an intricate shape based on volumetric input. Specifically, given a 3D brain MRI volume, we aim to reconstruct the highly folded white matter surface region. We first extract the initial surface from the white matter segmentation mask of the brain MRI image, then

we deform the initial surface to the target surface by modeling its vertices trajectory via neural ODE (Chen et al., 2018). Our deformation model is optimized via sliced Wasserstein distance loss by encoding the mesh as a probability measure. We further represent mesh as an oriented varifold and empirically show that our approach surpasses other related works on multiple datasets and metrics, namely, almost self-intersection-free while maintaining high geometric accuracy. It is worth noting that although our DDOT is developed within the scope of cortical surface reconstruction (CSR), the underlying concepts can be extended to other 3D deformation networks.

**Contribution.** In summary, our contributions are as follows:

1. We propose to represent triangle meshes as probability measures that generalize the common set-based approach in a learning-based deformation network. Specifically, we present three forms of mesh as probability measure: continuous, empirical, and discrete measure via an oriented varifold.

2. We propose a new metric for learning mesh deformation. Our metric utilizes the sliced Wasserstein distance (SWD), which operates on meshes represented as probability measures. We demonstrate that sliced Wasserstein distance (SWD) is a valid computationally fast metric between probability measures and provide the approximation bound between the SWD between empirical probability measures and the SWD between continuous probability measures.

3. We conduct extensive experiments on white matter reconstruction by employing neural ODE (Chen et al., 2018) to deform the initial surface to the target surface. Our experiments on multiple brain datasets demonstrate that our method outperforms existing state-of-the-art related works in terms of *geometric accuracy*, *self-intersection ratio*, and *consistency*.

**Organization.** The paper's structure is as follows. We first provide background about the set-based approach for comparing two meshes and the definition of Wasserstein distance as well as diffeomorphic flows for deforming meshes in Section 2. In Section 3, we propose probability measure encoding to represent mesh and further employ SWD as an objective function for diffeomorphic deformation framework as well as analyze their relevant theoretical properties. Section 4 presents our experiments on reconstructing cortical surface and provides quantitative results compared to state-of-the-art methods. In Section 5, we identify the limitations of our approach and outline potential avenues for future research. We then draw concluding remarks in Section 6. Finally, we defer the proofs of key results, supplementary materials, and discussion on related works to Appendices.

**Notations.** For any $d \geq 2$, we denote $\mathbb{S}^{d-1} := \{\theta \in \mathbb{R}^d \mid ||\theta||_2^2 = 1\}$ and $\mathcal{U}(\mathbb{S}^{d-1})$ as the unit hypersphere and its corresponding uniform distribution. We denote $\theta \sharp \mu$ as the push-forward measures of $\mu$ through the function $f : \mathbb{R}^d \to \mathbb{R}$ that is $f(x) = \theta^\top x$. Furthermore, we denote $\delta_x$ as Dirac distribution at a location $x$, and $\mathcal{P}_p(\mathbb{R}^d)$ is the set of probability measures over $\mathbb{R}^d$ that has finite $p$-moment. By abuse of notations, we use capitalized letters for both random variables and sets.

## 2 BACKGROUND

In this section, we first review the set-based approach for comparing two meshes. After that, we review the definition of Wasserstein distance and diffeomorphic flows for deforming meshes.

### 2.1 THE SET-BASED APPROACH: MESH TO POINT-CLOUD

**Mesh to a point-cloud.** To sample a point $p$ from a mesh, a face $f = (x_1, x_2, x_3)$ is first sampled with the probability proportional to the area of the face. Then, the position $p$ can be sampled by setting $p := w_1 x_1 + w_2 x_2 + w_3 x_3$, where $w_1 + w_2 + w_3 = 1$ are random barycentric coordinates which are uniformly distributed over a triangle (Ravi et al., 2020; Wang et al., 2019). The process is repeated until getting the desired number of points.

**Comparing two point-clouds.** After having representative point-clouds from meshes, a discrepancy in the space of point-clouds (sets) is used. The most widely used discrepancy for point-cloud is the set-based Chamfer pseudo distance (Barrow et al., 1977). For any two point-clouds $X$ and $Y$, the Chamfer distance is:

$$\text{CD}(X, Y) = \frac{1}{|X|} \sum_{x \in X} \min_{y \in Y} \|x - y\|_2^2 + \frac{1}{|Y|} \sum_{y \in Y} \min_{x \in X} \|x - y\|_2^2, \tag{1}$$

where $|X|$ denotes the number of points in $X$.

## 2.2 WASSERSTEIN DISTANCE

We now review the definition of the Wasserstein distance for comparing two probability measures $\mu \in \mathcal{P}_p(\mathbb{R}^d)$ and $\nu \in \mathcal{P}_p(\mathbb{R}^d)$. The Wasserstein-$p$ (Villani, 2003) distance between $\mu$ and $\nu$ as follows:

$$\mathrm{W}_p^p(\mu, \nu) := \inf_{\pi \in \Pi(\mu,\nu)} \int_{\mathbb{R}^d \times \mathbb{R}^d} \|x - y\|_p^p d\pi(x, y), \tag{2}$$

where $\Pi(\mu, \nu)$ is the set of joint distributions that have marginals are $\mu$ and $\nu$ respectively. A benefit of Wasserstein distance is that it can handle two measures that have disjointed supports.

**Wasserstein distance between continuous measures.** Computing the Wasserstein distance accurately between continuous measures is still an open question (Korotin et al., 2021) due to the non-optimality and instability of the minimax optimization for continuous functions which are the Kantorovich potentials (Arjovsky et al., 2017). Hence, discrete representations of the continuous measures are often used as proxies to compare them. i.e., the plug-in estimator (Bernton et al., 2019). In particular, let $X_1, \ldots, X_m \overset{i.i.d}{\sim} \mu$ and $Y_1, \ldots, Y_m \overset{i.i.d}{\sim} \nu$, the Wasserstein distance $W_p(\mu, \nu)$ is approximated by $W_p(\hat{\mu}_m, \hat{\nu}_m)$ with $\hat{\mu}_m = \frac{1}{m} \sum_{i=1}^m \delta_{X_i}$ and $\hat{\nu}_m = \frac{1}{m} \sum_{i=1}^m \delta_{Y_i}$ are the corresponding empirical measures. However, the convergence rate of the Wasserstein distance is $\mathcal{O}(m^{-1/d})$ (Mena & Weed, 2019). Namely, we have $\mathbb{E}\left[|W_p(\hat{\mu}_m, \hat{\nu}_m) - W_p(\mu, \nu)|\right] \leq Cm^{-1/d}$ for an universal constant $C$, where $\hat{\mu}_m = \frac{1}{m} \sum_{i=1}^m \delta_{X_i}$ is the corresponding empirical measure. Therefore, the Wasserstein distance suffers from the curse of dimensionality i.e., the Wasserstein distance needs more samples to represent the true measure well when dealing with high-dimensional measures. In the setting of comparing meshes, the Wasserstein distance will be worse if we use more features for meshes e.g., normals, colors, and so on.

**Wasserstein distance between discrete measures.** When $\mu$ and $\nu$ are two discrete probability measures that have at most $m$ supports, the time complexity and memory complexity to compute the Wasserstein distance are $\mathcal{O}(m^3 log m)$ and $\mathcal{O}(m^2)$ respectively. Therefore, using the plug-in estimator requires expensive computation since it requires a relative large value of $m$ to the dimension.

## 2.3 DIFFEOMORPHIC FLOWS

Diffeomorphic flows can be established by dense point correspondences between source and target surfaces. Given an input surface, the trajectories of the points can be modeled by an ODE, where the derivatives of the points are parameterized by a deep neural network (DNN). Specifically, let $\Phi(\boldsymbol{p}, t) : \Omega \subset \mathbb{R}^3 \times [0, 1] \mapsto \Omega \subset \mathbb{R}^3$ be a continuous hidden state of the neural network that defines a trajectory from source position $\boldsymbol{p} = \Phi(\boldsymbol{p}, 0)$ to the target position $\boldsymbol{p}' = \Phi(\boldsymbol{p}, 1)$, and $\mathbb{F}_\phi$ be a DNN with parameters $\phi$. An ordinary differential equation (ODE) with the initial condition is defined as:

$$\frac{\partial \Phi(\boldsymbol{p}, t)}{\partial t} = \mathbb{F}_\phi(\Phi(\boldsymbol{p}, t), t) \quad \text{s.t.} \quad \Phi(\boldsymbol{p}, 0) = \boldsymbol{p}, \tag{3}$$

If $\mathbb{F}_\phi$ is Lipschitz, a solution to Eq. 3 exists and is unique in the interval $[0, 1]$, which provides a theoretical guarantee that any two deformation trajectories do not intersect with each other (Coddington & Levinson, 1984).

## 3 DIFFEOMORPHIC MESH DEFORMATION VIA AN EFFICIENT OPTIMAL TRANSPORT METRIC

In this section, we generalize the set-based mesh representation by proposing three ways of transforming mesh into probability measure encoding. We further employ sliced Wasserstein distance as an objective function in the diffeomorphic flow model for cortical surface reconstruction task.

### 3.1 THE MEASURE-APPROACH: MESH TO PROBABILITY MEASURE

We now discuss the approach that we rely on to compare meshes via probability metrics. In particular, we consider transforming a mesh into a probability measure.

**Mesh to a continuous and hierarchical probability measure.** Let a mesh $\mathcal{M}$ have a set of faces $F^{\mathcal{M}} = \{f_1^{\mathcal{M}}, \ldots, f_N^{\mathcal{M}}\}$ ($N > 0$) where a face $f$ is represented by its vertices

$\text{Ver}(f) = \{x_1, \ldots, x_{v^f}\}$ ($v^f \geq 3$). We now can define a probability measure over faces, namely, $\mu^{\mathcal{M}}(f) = \sum_{i=1}^{N} \frac{\text{Vol}(f_i^{\mathcal{M}})}{\sum_{j=1}^{N} \text{Vol}(f_j^{\mathcal{M}})} \delta_{f_i^{\mathcal{M}}}$ is the categorical distribution over the faces that has the weights proportional to the areas of the faces (volume of the convex hull of vertices). For example, in the case of triangle meshes, we have $\text{Ver}(f) = (x_1, x_2, x_3)$ and $\text{Vol}(f) = \frac{1}{2}\|(x_2 - x_1) \times (x_3 - x_1)\|$. Given a face $f$, the conditional distribution for a point in the space is $\mu^{\mathcal{M}}(x|f) = \frac{1}{\text{Vol}(f)}, \forall x \in$ ConvexHull($\text{Ver}(f)$). Therefore, the marginal distribution for a point in the space induced by a mesh $\mathcal{M}$ is $\mu^{\mathcal{M}}(x) = \sum_{i=1}^{N} \mu^{\mathcal{M}}(x|f_i^{\mathcal{M}})\mu^{\mathcal{M}}(f_i^{\mathcal{M}})$. It is worth noting that given two meshes $\mathcal{M}_1$ and $\mathcal{M}_2$, their corresponding probability measures $\mu^{\mathcal{M}_1}$ and $\mu^{\mathcal{M}_2}$ are likely to have disjoint supports. Therefore, the optimal transport distances are natural metrics for comparing them.

**Mesh to an empirical probability measure.** As discussed in the background, the most computationally efficient and stable approach to approximate the Wasserstein distance is through the plugin estimator (Bernton et al., 2019). In particular, let $x_1, \ldots, x_m$ be a set of points that are independently identically sampled from $\mu^{\mathcal{M}_1}$ and $y_1, \ldots, y_m$ be a set of points that are independently identically sampled from $\mu^{\mathcal{M}_2}$. After that, we can define $\hat{\mu}_m^{\mathcal{M}_1} = \frac{1}{m} \sum_{i=1}^{m} \delta_{x_i}$ and $\hat{\mu}_m^{\mathcal{M}_2} = \frac{1}{m} \sum_{i=1}^{m} \delta_{y_i}$ as the represented empirical probability measure of the two meshes $\mathcal{M}_1$ and $\mathcal{M}_2$ respectively. Finally, the discrete Wasserstein distance is computed between $\hat{\mu}_m^{\mathcal{M}_1}$ and $\hat{\mu}_m^{\mathcal{M}_2}$ as the final discrepancy.

**Mesh to a discrete probability measure.** To have a richer representation of mesh, we consider the discrete probability measure representation through varifold. Let $M$ be a smooth submanifold of dimension 2 embedded in the ambient space of $\mathbb{R}^n$, e.g. $n = 3$ for surface, with finite total volume $\text{Vol}(M) < \infty$. For every point $p \in M$, there exists a tangent space $T_p M$ be a linear subspace of $\mathbb{R}^n$. To establish an orientation of $M$, it is essential to orient the tangent space $T_p M$ for every $p \in M$. This ensures that each oriented tangent space can be represented as an element of an oriented Grassmannian. Inspired from previous works (Glaunes et al., 2004; Vaillant & Glaunes, 2005; Charon & Trouvé, 2013; Kaltenmark et al., 2017), $M$ can be associated as an oriented varifold $\tilde{\mu}^M$, i.e. a distribution on the position space and tangent space orientation $\mathbb{R}^n \times \mathbb{S}^{n-1}$, as follows: $\tilde{\mu}^M = \int_M \delta_{(p, \vec{n}(p))} d\text{Vol}(p)$, where $\vec{n}(p)$ is the unit oriented normal vector to the surface at $p$. Once established the oriented varifold for a smooth surface, an oriented varifold for triangular mesh $\mathcal{M}$ that approximates smooth shape $M$ can be derived as follows:

$$\tilde{\mu}^{\mathcal{M}} = \sum_{i=1}^{|F|} \tilde{\mu}^{f_i} = \sum_{i=1}^{|F|} \int_{f_i} \delta_{(p_i, \vec{n}(p_i))} d\text{Vol}(p) \approx \sum_{i=1}^{|F|} \alpha_i \delta_{(p_i, \vec{n}(p_i))}, \tag{4}$$

where $p_i$ is the barycenter of the vertices of face $f_i$ and $\alpha_i := \text{Vol}(f_i)$ is the area of the triangle. To ensure that $\tilde{\mu}^{\mathcal{M}}$ possesses the characteristic of a discrete measure, we normalize $\alpha_i$s' such that they sum up to 1. Note that provided the area of triangular is sufficiently small, $\tilde{\mu}^{\mathcal{M}}$ gives an acceptable approximation of the discrete mesh $\mathcal{M}$ in terms of oriented varifold (Kaltenmark et al., 2017).

## 3.2 EFFECTIVE AND EFFICIENT MESH COMPARISON WITH SLICED WASSERSTEIN

**Sliced Wasserstein distance.** The sliced Wasserstein distance (Bonneel et al., 2015) (SWD) between two probability measures $\mu \in \mathcal{P}_p(\mathbb{R}^d)$ and $\nu \in \mathcal{P}_p(\mathbb{R}^d)$ is defined as:

$$\text{SW}_p^p(\mu, \nu) = \mathbb{E}_{\theta \sim \mathcal{U}(\mathbb{S}^{d-1})}[\text{W}_p^p(\theta \sharp \mu, \theta \sharp \nu)], \tag{5}$$

The benefit of SW is that $\text{W}_p^p(\theta \sharp \mu, \theta \sharp \nu)$ has a closed-form solution which is $\int_0^1 |F_{\theta \sharp \mu}^{-1}(z) - F_{\theta \sharp \nu}^{-1}(z)|^p dz$ with $F^{-1}$ denotes the inverse CDF function. The expectation is often approximated by Monte Carlo sampling, namely, it is replaced by the average from $\theta_1, \ldots, \theta_L$ ($L$ is the number of projections) that are drawn i.i.d from $\mathcal{U}(\mathbb{S}^{d-1})$. In particular, we have:

$$\widehat{SW}_p^p(\mu, \nu) = \frac{1}{L} \sum_{l=1}^{L} \text{W}_p^p(\theta_l \sharp \mu, \theta_l \sharp \nu). \tag{6}$$

The computational complexity and memory complexity of the Monte Carlo estimation of SW are $\mathcal{O}(Lm \log m)$ and $\mathcal{O}(Lm)$ (Nguyen & Ho, 2024) respectively when $\mu$ and $\nu$ are discrete measures with at most $m$ supports. Therefore, the SW is naturally suitable for large-scale mesh comparison.

**Convergence rate.** We now discuss the non-asymptotic convergence of the Monte Carlo estimation of the sliced Wasserstein between two empirical probability measures to the sliced Wasserstein between the corresponding two continuous probability measures on surface meshes.

**Theorem 1.** *For any two meshes $\mathcal{M}_1$ and $\mathcal{M}_2$, let $X_1, \ldots, X_m \overset{i.i.d}{\sim} \mu^{\mathcal{M}_1}(x)$, $Y_1, \ldots, Y_m \overset{i.i.d}{\sim} \mu^{\mathcal{M}_2}(x)$, $\hat{\mu}_m^{\mathcal{M}_1}(x) = \frac{1}{m} \sum_{i=1}^m \delta_{X_i}$ and $\hat{\mu}_m^{\mathcal{M}_2}(x) = \frac{1}{m} \sum_{i=1}^m \delta_{Y_i}$ be the corresponding empirical distribution. Assume that $\mu^{\mathcal{M}_1}$ and $\mu^{\mathcal{M}_2}$ have compact supports with the diameters that are at most $R$, we have the following approximation error :*

$$
\mathbb{E}\left[\left|\widehat{SW}_p^p(\hat{\mu}_m^{\mathcal{M}_1}, \hat{\mu}_m^{\mathcal{M}_2}; L) - SW_p^p(\mu^{\mathcal{M}_1}, \mu^{\mathcal{M}_2})\right|\right] \leq RC_{p,R}\sqrt{\frac{(d+1)\log m}{m}}
$$
$$
+ \frac{1}{\sqrt{L}}\mathbb{E}\left[Var\left[W_p^p(\theta\sharp\hat{\mu}_m^{\mathcal{M}_1}, \theta\sharp\hat{\mu}_m^{\mathcal{M}_2})\right]^{1/2}|X_{1:m}, Y_{1:m}\right],
$$

*for an universal constant $C_{p,R} > 0$. The variance is with respect to $\theta \sim \mathcal{U}(\mathbb{S}^{d-1})$.*

Theorem 1 suggests that when using empirical probability measure to approximate meshes, the error between the Monte Carlo estimation of sliced Wasserstein distance between empirical distributions over two sampled point-clouds and sliced Wasserstein distance between two continuous distributions on meshes surface is bounded by the rate of $m^{-1/2}$ and $L^{-1/2}$. It means that, when increasing the number of points and the number of projections, the error reduces by the square root of them. This rate is very fast since it does not depend exponentially on the dimension, hence, it is scalable to meshes with high-dimensional features at vertices, such as normals, colors, and so on. Leveraging the scaling property and the approximation of varifold to mesh $\mathcal{M}$ mentioned in Sec 3.1, we can represent meshes as discrete measures $\tilde{\mu}^{\mathcal{M}}$ and optimize $\widehat{SW}_p^p(\tilde{\mu}^{\mathcal{M}_1}, \tilde{\mu}^{\mathcal{M}_2}; L)$ as the objective function. The proof of Theorem 1 is given in Appendix C. It is worth noting that a similar property is not able to be derived for Chamfer since it is a discrepancy on sets that cannot be generalized to compare meshes.

### 3.3 SLICED WASSERSTEIN DIFFEOMORPHIC FLOW FOR CORTICAL SURFACE RECONSTRUCTION

**Diffeomorphic deformation framework for reconstructing cortical surfaces.** In this section, we present DDOT that incorporates diffeomorphic deformation and sliced Wasserstein distance to reconstruct the cortical surface. Specifically, our goal is to derive a high-resolution, 2D manifold of the white matter that is topologically accurate from a 3D brain MR image. Let $\mathbf{I} \in \mathbb{R}^{D \times W \times H}$ be a MRI volume and $\mathcal{M} = (\mathcal{V}, \mathcal{F})$ be a 3D triangle mesh. The corresponding vertices of the mesh are represented by $v \in \mathbb{R}^3$. Firstly, we train a U-Net model to automatically predict the white matter segmentation mask from $\mathbf{I}$. Then, a signed distance function (SDF) is extracted from the binary mask before employing Marching Cubes (Lorensen & Cline, 1987) to get the initial surface. After getting the initial surface $\mathcal{M}_0$, the trajectory of each coordinate $v_0$ is modeled via the ODE with initial condition from Eq. 3. Inspired from (Ma et al., 2021; 2022), we concatenate the point features with the corresponding cube features sampling from $\mathbf{I}$ as a new feature vector. The new feature is passed through a multilayer perceptron $\mathbb{F}_\phi$ to learn the deformation. More implementation settings are included in the Appendix D.

**Sliced Wasserstein distance as a loss function.** To train the DDOT model, we minimize the distance between predicted mesh $\hat{\mathcal{M}}$ and the ground truth mesh $\mathcal{M}^*$. We adopt a novel way of transforming mesh into a probability measure and leveraging sliced Wasserstein distance (SWD) as a loss function to optimize two discrete meshes. As discussed, the SW is a valid metric on the space of distribution and can guarantee the convergence of the probability measure. Moreover, as shown in Theorem 1, the sample complexity of the SW is bounded with a parametric rate, hence, it is suitable to use the SW to compare empirical probability measures as the proxy for the continuous mesh probability measure. Therefore, we sample points on $\hat{\mathcal{M}}$ and $\mathcal{M}^*$ as probability measures and compute SWD loss between these two measures without regularization terms. Additionally, we can represent mesh as discrete probability measures, i.e. oriented varifold, and utilize the same objective function. Based on our observations, the varifold representation has shown better performance compared to encoding using empirical probability measures, which we provide a more detailed comparison in our ablation study in Sec. 4.3. As a result, we assume that our experiments in the following sections will be conducted using the oriented varifold approach unless otherwise specified.

Table 1: **Quantitative results of white matter surface reconstruction** in terms of earth mover's distance (EMD), sliced Wasserstein distance (SWD), average symmetric surface distance (ASSD), Chamfer normals (CN), and self-intersection face ratio (SI) on ADNI and OASIS datasets. Best values are highlighted. EMD, SWD, ASSD results are in mm. All results are listed in the format "mean value $\pm$ standard deviation". **DDOT** represents the reconstruction results from our proposed approach. While $\downarrow$ means smaller metric value is better, $\uparrow$ indicates a larger metric value is better.

| ADNI dataset | Left WM | | | | | Right WM | | | | |
|---|---|---|---|---|---|---|---|---|---|---|
| Method | EMD (mm) ↓ | SWD (mm) ↓ | ASSD (mm) ↓ | CN ↑ | SI (%) ↓ | EMD (mm) ↓ | SWD (mm) ↓ | ASSD (mm) ↓ | CN ↑ | SI (%) ↓ |
| DeepCSR | 1.368 ±.721 | 1.357 ±1.178 | .390 ±.162 | .934 ±.016 | \ | 1.350 ±.350 | 1.357 ±.589 | .388 ±.172 | .936 ±.014 | \ |
| Vox2Cortex | 1.051 ±.173 | .823 ±.351 | .346 ±.073 | .926 ±.011 | .719 ±.214 | 1.048 ±.134 | .811 ±.294 | .335 ±.061 | .927 ±.010 | .745 ±.199 |
| CFPP | .912 ±.435 | .525 ±.265 | .271 ±.071 | .936 ±.009 | .058 ±.032 | .821 ±.169 | .473 ±.286 | .268 ±.073 | .933 ±.009 | .067 ±.032 |
| CortexODE | .803 ±.136 | .436 ±.403 | .234 ±.064 | .938 ±.010 | .013 ±.011 | .782 ±.081 | .384 ±.259 | .231 ±.052 | **.939** ±.019 | .004 ±.005 |
| Ours | **.728** ±.013 | **.420** ±.273 | **.202** ±.043 | **.945** ±.012 | $< 10^{-4}$ | **.702** ±.068 | **.365** ±.223 | **.205** ±.056 | .938 ±.012 | $< 10^{-4}$ |

| OASIS dataset | Left WM | | | | | Right WM | | | | |
|---|---|---|---|---|---|---|---|---|---|---|
| Method | EMD (mm) ↓ | SWD (mm) ↓ | ASSD (mm) ↓ | CN ↑ | SI (%) ↓ | EMD (mm) ↓ | SWD (mm) ↓ | ASSD (mm) ↓ | CN ↑ | SI (%) ↓ |
| DeepCSR | .887 ±.787 | 1.020 ±.392 | .312 ±.124 | .941 ±.010 | \ | .900 ±.740 | 1.072 ±.419 | .344 ±.158 | .941 ±.011 | \ |
| Vox2Cortex | .594 ±.236 | .876 ±.053 | .302 ±.037 | .928 ±.008 | .994 ±.193 | .574 ±.256 | .872 ±.062 | .303 ±.042 | .929 ±.009 | 1.022 ±.186 |
| CFPP | .511 ±.222 | .841 ±.059 | .225 ±.038 | .937 ±.007 | .054 ±.060 | .473 ±.224 | .840 ±.071 | .227 ±.046 | .935 ±.008 | .076 ±.068 |
| CortexODE | .425 ±.193 | .785 ±.047 | .183 ±.036 | .943 ±.007 | .032 ±.025 | .434 ±.256 | .787 ±.065 | .182 ±.052 | .943 ±.008 | .022 ±.020 |
| Ours | **.418** ±.192 | **.779** ±.055 | **.161** ±.040 | **.949** ±.005 | $< 10^{-4}$ | .429 ±.250 | **.770** ±.059 | **.160** ±.046 | **.949** ±.008 | $< 10^{-4}$ |

## 4 EXPERIMENTS

Within this section, we first provide detailed settings and conduct extensive experiments on multiple datasets. Furthermore, we also give a comprehensive ablation study to validate our findings.

### 4.1 EXPERIMENTAL SETTINGS

**Datasets.** We conduct experiments on three publicly available datasets: ADNI dataset (Jack Jr et al., 2008), OASIS dataset (Marcus et al., 2007), and TRT dataset (Maclaren et al., 2014). The pseudo-ground truth surfaces are obtained from Freesurfer v5.3 (Fischl, 2012). We want to emphasize that using pseudo-ground truth as a reference is a standard practice in brain image analysis, adopted by all of the methods we have included in our comparison (Cruz et al., 2021; Bongratz et al., 2022; Santa Cruz et al., 2022; Ma et al., 2022). Therefore, using pseudo-ground truth does not limit the significance of our contributions within the context of this paper. Regarding data split, we carefully stratified (at the patient-level) each dataset into train, valid, and test sets. We select the best checkpoint based on train and validation set, subsequently reporting the outcomes on the unseen test set. Additional dataset details are available in Appendix E.

**Baselines.** We reproduce all competing methods using their official implementations and recommended experimental settings based on our split for a fair comparison. Specifically, we reproduce DeepCSR (Cruz et al., 2021) in both the occupancy field and signed distance function (SDF) and report SDF results due to better performance. For Vox2Cortex (Bongratz et al., 2022), we use the authors' suggestion with a high-resolution template with $\approx 168,000$ vertices for each structure to get the best performance. Regarding CorticalFlow (Lebrat et al., 2021), we reproduce with their improved version settings CFPP (Santa Cruz et al., 2022). Finally, we retrain CortexODE (Ma et al., 2022) with default settings.

**Metrics.** We employ various metrics including earth mover's distance (EMD), sliced Wasserstein distance (SWD), average symmetric surface distance (ASSD), Chamfer normals (CN), and self-intersection faces ratio (SI). We sample 100K points over the predicted and target surface to compute EMD, SWD, ASSD, and CN. Due to the large number of sampled points, we estimate EMD using entropic regularization and the Sinkhorn algorithm from (Feydy et al., 2019). Furthermore, we determine SI faces using PyMesh (Zhou, 2019) library.

### 4.2 RESULTS & DISCUSSION

#### 4.2.1 RESULTS

**Geometric accuracy.** As shown in Tab. 1, our DDOT provides more geometrically accurate surfaces than other competing methods in multiple metrics. Qualitative results from Fig. 2 also indicate that our proposed method is closer to the ground truth than competing methods.

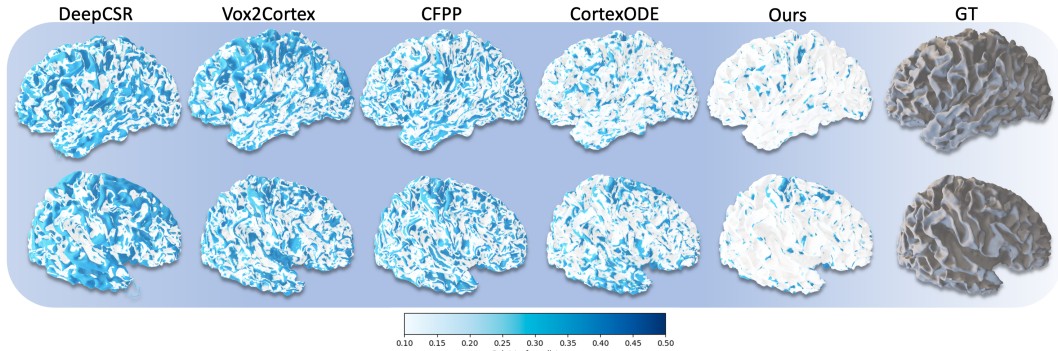

Figure 2: **Qualitative results of white matter surface reconstruction.** The color represents the point-to-face distance, i.e., the darker color is, the further the predicted mesh to the pseudo-ground truth. More visualization is given in Appendix F and the video supplementary.

**Self-intersection.** Compared to GNN deformation-based methods such as Vox2Cortex with no SI-free theoretical guarantee, diffeomorphic deformation-based methods such as CFPP, CortexODE, and our proposed approach has much less SI. However, despite the nice property of existence and uniqueness of the solution of ODE models, CFPP and CortexODE still introduce a certain amount of SI faces since they both rely on the optimization of Chamfer divergence on discretized vertices of the mesh. Our approach, on the other hand, represents mesh as probability measures and has strong theoretical optimization support by employing efficient optimal transport metric, thus can approximate the mesh much better with almost no SI faces, i.e. less than $10^{-4}\%$. It is worth noting that DDOT is $100\times$ better in terms of SIF score compared to CortexODE, the current SOTA in CSR task. DeepCSR introduces no SI thanks to Marching Cubes (Lorensen & Cline, 1987) from the implicit surface but often has other artifacts and requires extensive topology correction.

**Consistency.** We compare the consistency of our DDOT, Vox2Cortex, CortexODE (which are all trained on OA-SIS), and FreeSurfer on the TRT dataset. We reconstruct white matter cortical surfaces from MRI images of the same subject on the same day and evaluated the EMD, SWD, and ASSD of the resulting reconstructions. The expectation is that the brain morphology of two consecutive scans taken on the same day should be similar to each other, except for the variations caused by the imaging process. To align pairs of images, we utilized the iterative closest-point algorithm (ICP) following (Cruz et al., 2021). As presented in Tab. 2, we outperform in both EMD and ASSD, and only Freesurfer (Fischl, 2012) result has the better performance in SWD score than us.

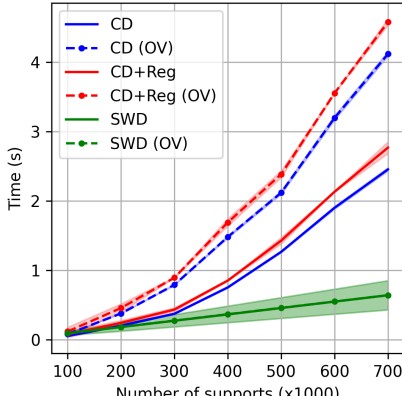

Figure 3: **Running time comparison.** The diagram indicates the scalability of three presented losses. The lines imply the losses computed between two sets of points in 3D-coordinate, the dashed lines with dots represent the loss computed between two varifolds. *OV* denotes oriented varifold, *Reg* denotes regularization.

### 4.2.2 RUNNING TIME ANALYSIS

We compare the running time of CD loss, CD loss with regularization, and SWD loss, as shown in Fig. 3. The regularization of CD loss includes mesh edge loss, normal consistency, and Laplacian smoothing, which are commonly employed in mesh deformation frameworks (Bongratz et al., 2022; Santa Cruz et al., 2022). Firstly, as the number of supports increases, SWD loss consistently exhibits significantly faster performance compared to CD losses. This empirical finding further substantiates our assertion regarding the theoretical running time complexities of SWD ($\mathcal{O}(m \log m)$) and CD ($\mathcal{O}(m^2)$), with $m$ denotes the number of supports. Secondly, regarding high-dimensional measures, e.g. varifolds, while the CD losses rigorously scale w.r.t. the dimension of the supports, SWD loss shows minimal variation as the number of supports increases, further support Theorem 1.

In conclusion, our proposed metric is scalable w.r.t. both the number of supports and the dimension of those supports, thus demonstrating the efficiency of our methods in learning-based frameworks.

## 4.3 ABLATION STUDY

**Setups.** We evaluate the individual optimization design choices and report the WM surface reconstruction on ADNI dataset. For fair comparisons, we conduct ablation studies on the same initial surface and the same number of supports, i.e. the number of faces on mesh. We train all of them for 300 epochs and get the best checkpoints on the validation set. The result is reported in Tab. 3 on the holdout test set.

**Comparisons.** To begin with, we conduct experiments where we independently identically sample points on the surface and used SWD loss to optimize two sets of points. The results showed that this approach did not perform as well as optimizing SWD on oriented varifold representation. This indicates that the varifold method provides better mesh approximation compared to using random points on the surface. In the second part of our ablations, we use CD to optimize two oriented varifolds. Our results show that SWD loss outperforms CD on varifold, which further supports our Theorem 1. Finally, we employ the Sinkhorn divergence, implemented by (Séjourné et al., 2019), as the loss function to optimize two oriented varifolds. It is worth noting that Sinkhorn divergence is the approximation of Wasserstein distance, but cannot lead to a valid metric between probability measures since the resulting discrepancy does not satisfy the triangle inequality. Experiments on both left and right WM show that our SWD loss outperforms Sinkhorn in both metrics.

Table 2: **White matter surface reconstruction consistency comparison** in terms of EMD, SWD, ASSD on TRT dataset.

| Method | EMD | SWD | ASSD |
|---|---|---|---|
| Vox2Cortex | .886 ±.130 | .485 ±.176 | .263 ±.112 |
| CortexODE | .799 ±.038 | .444 ±.201 | .241 ±.040 |
| FreeSurfer | .859 ±.213 | **.358** ±.275 | .286 ±.156 |
| Ours | **.780** ±.032 | .403 ±.184 | **.229** ±.010 |

## 5 LIMITATIONS AND FUTURE WORKS

Our work is the first learning-based deformation approach that tackles the local optimality problem of Chamfer distance on mesh deformation by employing efficient optimal transport theory on meshes as probability measures. Yet, it is not without its limitations, which present intriguing avenues for future exploration. Unlike the set-based approach that predefines the number of sampling supports, our optimization settings work best on the deterministic supports correlated with the mesh resolution, thus introducing stochastic memory during training. Our future work will focus on mitigating this issue by either employing remeshing techniques or ensuring a consistent cutoff number of supports for both the predicted mesh and the target mesh. Secondly, though the underlying proposed techniques have potential applications in other deformation tasks beyond CSR, within the context of this paper, we only focus on this task. It is intriguing to explore the potential applications of our approach in diverse domains.

Table 3: **Ablation Study** in terms of EMD and SWD metric. *P.S.* and *O.V.* are short for *point sampling* and *oriented varifold* representation, respectively. Best values are highlighted.

| | Left WM | | Right WM | |
|---|---|---|---|---|
| | EMD | SWD | EMD | SWD |
| SWD on P.S. | .850 ±.134 | .450 ±.132 | .832 ±.052 | .420 ±.123 |
| CD on O.V. | .874 ±.121 | .499 ±.343 | .848 ±.077 | .529 ±.209 |
| Sinkhorn on O.V. | 1.023 ±.109 | .578 ±.307 | 1.002 ±.071 | .568 ±.178 |
| SWD on O.V. | **.728** ±.013 | **.420** ±.273 | **.702** ±.068 | **.365** ±.232 |

## 6 CONCLUSION

In this paper, we introduce a learning-based diffeomorphic deformation network that employs sliced Wasserstein distance (SWD) as the objective function to deform an initial mesh to an intricate mesh based on volumetric input. Different from previous approaches that use point-clouds for approximating mesh, we represent a mesh as a probability measure that generalizes the common set-based methods. By lying on probability measure space, we can further exploit statistical shape analysis theory to approximate mesh as an oriented varifold. Our theorem shows that leveraging sliced Wasserstein distance to optimize probability measures can have a fast statistical rate for approximating the surfaces of the meshes. Finally, we extensively verify our proposed approach in the challenging brain cortical surface reconstruction problem. Our experiment results demonstrate that our method surpasses existing state-of-the-art competing works in terms of geometric accuracy, self-intersection ratio, and consistency.

## 7 ACKNOWLEDGEMENTS

We thank Xiangyi Yan and Pooya Khosravi for their feedbacks on the paper. NH acknowledges support from the NSF IFML 2019844 and the NSF AI Institute for Foundations of Machine Learning.

## 8 ETHICS STATEMENT

**Potential impacts.** Our work experiments on the human brain MRI dataset. Our proposed model holds the potential to assist neuroradiologists in effectively visualizing brain surfaces. However, it is crucial to emphasize that our predictions should not be utilized for making clinical decisions. This is because our model has solely undergone testing using the data discussed within this research, and we cannot ensure its performance on unseen data in clinical practices.

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

# Supplement to "Diffeomorphic Mesh Deformation via Efficient Optimal Transport for Cortical Surface Reconstruction"

In this supplementary, we first discuss some related works in Appendix A. Then, we provide additional materials in Appendix B including the detailed setup and more insight about our toy example mentioned in Fig. 1. Secondly, we demonstrate complete proof for the Theorem 1 in Appendix C. Next, we delve into the implementation details in Appendix D and provide detailed information about the datasets in Appendix E. Finally, we present additional visualization of our experiment results in Appendix F.

## A   RELATED WORKS

**Deformation network for surface reconstruction.** 3D surface reconstruction can be obtained from various approaches such as volumetric, implicit surfaces, and geometric deep learning methods. While volumetric-based (Choy et al., 2016; Häne et al., 2017; Tatarchenko et al., 2017; Wang et al., 2018b; Cruz et al., 2021) and implicit surface-based (Mescheder et al., 2019; Park et al., 2019; Xu et al., 2019) methods can directly obtain surface by employing iso-surface extraction methods, such as Marching Cubes (Lorensen & Cline, 1987), they often require extensive post-processing to capture the high-quality resulting mesh. In contrast, geometric deep learning approaches use mesh deformation to achieve the target mesh while maintaining vertex connectivity (Pan et al., 2019; Smith et al., 2019; Wang et al., 2018a; Wickramasinghe et al., 2020; Nguyen et al., 2024b; Gupta, 2020; Bongratz et al., 2022). Among deformation-based approaches, diffeomorphic deformation demonstrates its capability to perform well on complex manifolds while keeping the '*manifoldness*' property (Gupta, 2020; Ma et al., 2022; Lebrat et al., 2021). However, those methods often use Chamfer divergence as their objective optimization, which is sub-optimal, especially on intricate manifolds such as cortical surfaces, i.e. as illustrated in Fig. 4. Therefore, in this work, we address the problem by employing efficient optimal transport in optimizing mesh during training diffeomorphic deformation models.

**Mesh as varifold representation.** Varifolds were initially introduced in the realm of geometric measure theory as a practical approach to tackle Plateau's problem (Almgren, 1966), which involves determining surfaces with a specified boundary that has the least area. Specifically, varifolds provide a convenient representation of geometric shapes, including rectifiable curves and surfaces, and serve as an effective geometric measure for optimization-based shape matching problems (Charon & Trouvé, 2013; Charon, 2013; Kaltenmark et al., 2017; Hsieh & Charon, 2020; Rekik et al., 2016; Ma et al., 2010). In this work, we focus on employing varifold as a discrete measure approximating the mesh. To the best of our knowledge, we are the first to exploit oriented varifolds as discrete probability measures in the learning-based deformation framework.

## B   TOY EXAMPLES

**Setups.** In this toy example, we aim to deform the template circle to the target polygon in an optimization-based. We uniformly sample the template circle and the target polygon into 2D points. The number of sampled points on both the template circle and target polygon are 678 points. To optimize the position of predicted points and target points, we employ Chamfer loss implemented by (Ravi et al., 2020), and the sliced Wasserstein distance with $p = 2$ approximated by Monte Carlo estimation with 100 projections. We optimize the two sets of points with stochastic gradient descent (SGD) optimizer with a learning rate of $1.0$ and momentum of $0.9$ for 1000 iterations.

**Discussion.** As shown in Fig. 5, we can see that the set of points optimized by Chamfer distance often gets trapped in some specific region, e.g. the acute region of the polygon in this example. This confinement occurs due to the nature of Chamfer distance, which primarily focuses on optimizing nearest neighbors, inhibiting the points from escaping the local region during the optimization process. To alleviate this issue, practitioners often introduce multiple losses as regularizers to aid Chamfer distance in escaping local minima. However, determining the appropriate weights for each auxiliary loss is a challenging task, as they tend to vary across different tasks, thus making the optimization process harder. SWD loss, on the other hand, can find the optimal transport plan for the whole set of points, thus resulting in a better solution when compared to Chamfer distance.

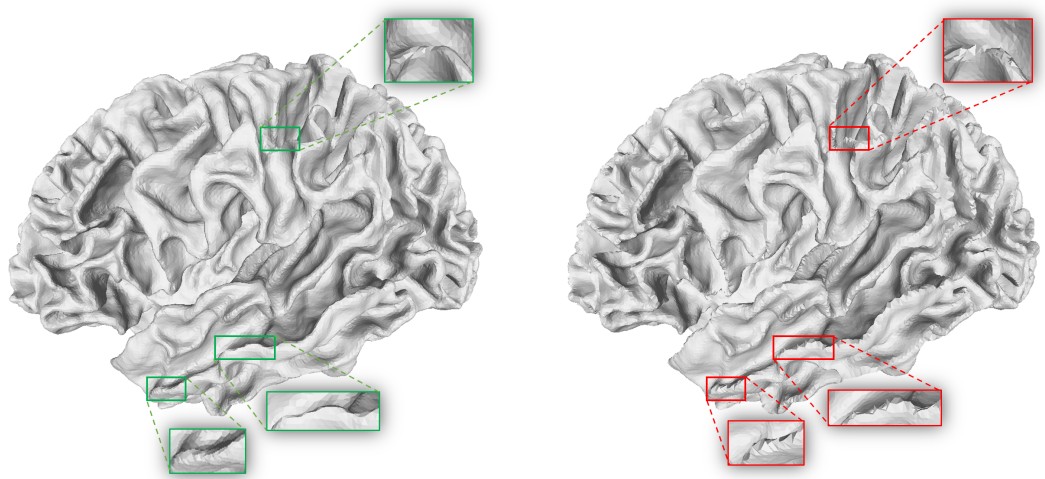

Figure 4: **Comparison between SWD loss (left) and CD loss (right).** The mesh obtained through probability measure representation and SWD optimization exhibits a more uniformly surfaced appearance compared to the set-based approach that optimizes with CD loss.

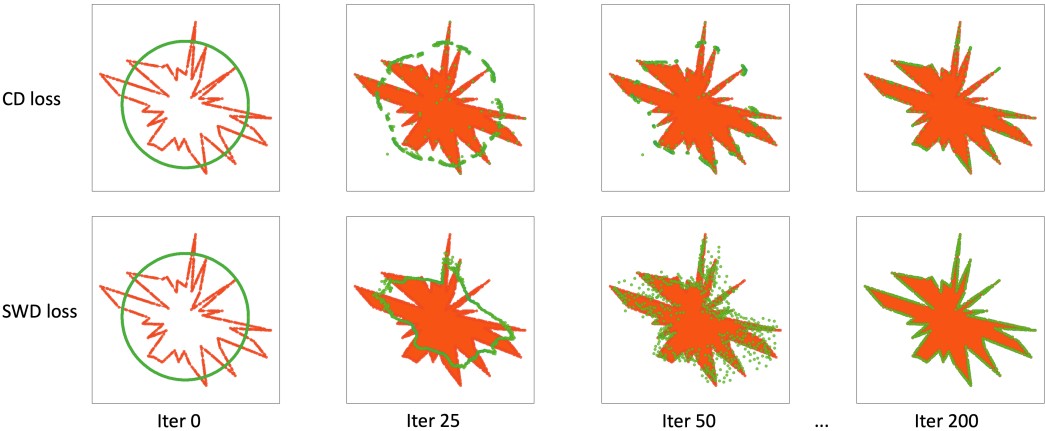

Figure 5: **Visualization of the optimization process of 2D toy example.** The set of green points, i.e. sampled points from the template circle, optimized by CD loss often concentrates around the acute region of the polygon and easily gets trapped at some local regions. Nonetheless, the set of points optimized by SWD loss distributes more uniformly along the edge of the polygon, thus making the optimization process more robust.

## C    PROOF OF THEOREM 1

Using the triangle inequality of the $\mathbb{L}_1$ norm, we obtain:

$$\mathbb{E}\left[\left|\widehat{\mathrm{SW}}_p^p(\hat{\mu}_m^{\mathcal{M}_1}, \hat{\mu}_m^{\mathcal{M}_2}; L) - \mathrm{SW}_p^p(\mu^{\mathcal{M}_1}, \mu^{\mathcal{M}_2})\right|\right] \leq \mathbb{E}\left[\left|\widehat{\mathrm{SW}}_p^p(\hat{\mu}_m^{\mathcal{M}_1}, \hat{\mu}_m^{\mathcal{M}_2}; L) - \mathrm{SW}_p^p(\hat{\mu}_m^{\mathcal{M}_1}, \hat{\mu}_m^{\mathcal{M}_2})\right|\right] \\ + \mathbb{E}\left[\left|\mathrm{SW}_p^p(\hat{\mu}_m^{\mathcal{M}_1}, \hat{\mu}_m^{\mathcal{M}_2}) - \mathrm{SW}_p^p(\mu^{\mathcal{M}_1}, \mu^{\mathcal{M}_2})\right|\right].$$

Now, we bound the first term $\mathbb{E}\left[\left|\widehat{SW}_p^p(\hat{\mu}_m^{\mathcal{M}_1}, \hat{\mu}_m^{\mathcal{M}_2}; L) - SW_p^p(\hat{\mu}_m^{\mathcal{M}_1}, \hat{\mu}_m^{\mathcal{M}_2})\right|\right]$. By the definitions of the SW and its Monte Carlo estimation, we have:

$$
\begin{aligned}
\mathbb{E}&\left[\left|\widehat{SW}_p^p(\hat{\mu}_m^{\mathcal{M}_1}, \hat{\mu}_m^{\mathcal{M}_2}; L) - SW_p^p(\hat{\mu}_m^{\mathcal{M}_1}, \hat{\mu}_m^{\mathcal{M}_2})\right|\right] \\
&= \mathbb{E}\left[\left|\frac{1}{L}\sum_{l=1}^L W_p^p(\theta_l\sharp\hat{\mu}_m^{\mathcal{M}_1}, \theta_l\sharp\hat{\mu}_m^{\mathcal{M}_2}) - \mathbb{E}[W_p^p(\theta\sharp\hat{\mu}_m^{\mathcal{M}_1}, \theta\sharp\hat{\mu}_m^{\mathcal{M}_2})|X_{1:m}, Y_{1:m}]\right|\right] \\
&\leq \left(\mathbb{E}\left[\left(\frac{1}{L}\sum_{l=1}^L W_p^p(\theta_l\sharp\hat{\mu}_m^{\mathcal{M}_1}, \theta_l\sharp\hat{\mu}_m^{\mathcal{M}_2}) - \mathbb{E}[W_p^p(\theta\sharp\hat{\mu}_m^{\mathcal{M}_1}, \theta\sharp\hat{\mu}_m^{\mathcal{M}_2})|X_{1:m}, Y_{1:m}]\right)^2\right]\right)^{\frac{1}{2}} \\
&= \left(\mathbb{E}\left[\left(\frac{1}{L}\sum_{l=1}^L \left(W_p^p(\theta_l\sharp\hat{\mu}_m^{\mathcal{M}_1}, \theta_l\sharp\hat{\mu}_m^{\mathcal{M}_2}) - \mathbb{E}[W_p^p(\theta\sharp\hat{\mu}_m^{\mathcal{M}_1}, \theta\sharp\hat{\mu}_m^{\mathcal{M}_2})|X_{1:m}, Y_{1:m}]\right)\right)^2\right]\right)^{\frac{1}{2}},
\end{aligned}
$$

where the inequality is due to the Holder's inequality. Using the fact that $\mathbb{E}\left[\frac{1}{L}\sum_{l=1}^L W_p^p(\theta_l\sharp\hat{\mu}_m^{\mathcal{M}_1}, \theta_l\sharp\hat{\mu}_m^{\mathcal{M}_2}|X_{1:m}, Y_{1:m})\right] = \mathbb{E}[W_p^p(\theta\sharp\hat{\mu}_m^{\mathcal{M}_1}, \theta\sharp\hat{\mu}_m^{\mathcal{M}_2})|X_{1:m}, Y_{1:m}]$ since $\theta_1, \ldots, \theta_L \overset{i.i.d}{\sim} \mathcal{U}(\mathbb{S}^{d-1})$, we have:

$$
\begin{aligned}
\mathbb{E}\left[\left|\widehat{SW}_p^p(\hat{\mu}_m^{\mathcal{M}_1}, \hat{\mu}_m^{\mathcal{M}_2}; L) - SW_p^p(\hat{\mu}_m^{\mathcal{M}_1}, \hat{\mu}_m^{\mathcal{M}_2})\right|\right] &\leq \left(\text{Var}\left[\left(\frac{1}{L}\sum_{l=1}^L W_p^p(\theta_l\sharp\hat{\mu}_m^{\mathcal{M}_1}, \theta_l\sharp\hat{\mu}_m^{\mathcal{M}_2})|X_{1:m}, Y_{1:m}\right)\right]\right)^{\frac{1}{2}} \\
&= \frac{1}{\sqrt{L}}\text{Var}\left[W_p^p(\theta\sharp\hat{\mu}_m^{\mathcal{M}_1}, \theta\sharp\hat{\mu}_m^{\mathcal{M}_2})|X_{1:m}, Y_{1:m}\right]^{\frac{1}{2}}.
\end{aligned}
$$

Now, we bound the second term $\mathbb{E}\left[\left|SW_p^p(\hat{\mu}_m^{\mathcal{M}_1}, \hat{\mu}_m^{\mathcal{M}_2}) - SW_p^p(\mu^{\mathcal{M}_1}, \mu^{\mathcal{M}_2})\right|\right]$. Using the Jensen inequality, we obtain

$$
\begin{aligned}
\mathbb{E}\left[\left|SW_p^p(\hat{\mu}_m^{\mathcal{M}_1}, \hat{\mu}_m^{\mathcal{M}_2}) - SW_p^p(\mu^{\mathcal{M}_1}, \mu^{\mathcal{M}_2})\right|\right] &= \mathbb{E}\left[\left|\mathbb{E}[W_p^p(\theta\sharp\hat{\mu}_m^{\mathcal{M}_1}, \theta\sharp\hat{\mu}_m^{\mathcal{M}_2})] - \mathbb{E}[W_p^p(\theta\sharp\mu^{\mathcal{M}_1}, \theta\sharp\mu^{\mathcal{M}_2})]\right|\right] \\
&\leq \mathbb{E}\left[\left|W_p^p(\theta\sharp\hat{\mu}_m^{\mathcal{M}_1}, \theta\sharp\hat{\mu}_m^{\mathcal{M}_2}) - W_p^p(\theta\sharp\mu^{\mathcal{M}_1}, \theta\sharp\mu^{\mathcal{M}_2})\right|\right] \\
&\leq C_{p,R}^{(1)}\mathbb{E}[|W_1(\theta\sharp\hat{\mu}_m^{\mathcal{M}_1}, \theta\sharp\mu^{\mathcal{M}_1}) + W_1(\theta\sharp\hat{\mu}_m^{\mathcal{M}_2}, \theta\sharp\mu^{\mathcal{M}_2})|] \\
&= C_{p,R}^{(1)}(\mathbb{E}[W_1(\theta\sharp\hat{\mu}_m^{\mathcal{M}_1}, \theta\sharp\mu^{\mathcal{M}_1})] + \mathbb{E}[W_1(\theta\sharp\hat{\mu}_m^{\mathcal{M}_2}, \theta\sharp\mu^{\mathcal{M}_2})]),
\end{aligned}
$$

where the second inequality is due to Lemma 4 in (Goldfeld et al., 2022). We now show that:

$$
\mathbb{E}[W_1(\theta\sharp\hat{\mu}_m^{\mathcal{M}_1}, \theta\sharp\mu^{\mathcal{M}_1})] \leq C_R^{(2)}\sqrt{\frac{(d+1)\log m}{m}}.
$$

Following (Nguyen et al., 2021a; Nguyen & Ho, 2023), we have:

$$
\begin{aligned}
\mathbb{E}[W_1(\theta\sharp\hat{\mu}_m^{\mathcal{M}_1}, \theta\sharp\mu^{\mathcal{M}_1})] &\leq \mathbb{E}\left[\max_{\theta\in\mathbb{S}^{d-1}} W_1(\theta\sharp\hat{\mu}_m^{\mathcal{M}_1}, \theta\sharp\mu^{\mathcal{M}_1})|X_{1:m}, Y_{1:m}\right] \\
&= \mathbb{E}\left[\max_{\theta\in\mathbb{R}^d|\|\theta\|_2\leq 1} W_1(\theta\sharp\hat{\mu}_m^{\mathcal{M}_1}, \theta\sharp\mu^{\mathcal{M}_1})|X_{1:m}, Y_{1:m}\right] \\
&= \mathbb{E}\left[\max_{\theta\in\mathbb{R}^d|\|\theta\|_2\leq 1} \int_0^1 |F_{m,\theta}^{-1}(z) - F_\theta^{-1}(z)|dz|X_{1:m}, Y_{1:m}\right],
\end{aligned}
$$

where $F_{m,\theta}^{-1}(z)$ is the inverse CDF of $\hat{\mu}_m^{\mathcal{M}_1}$, $F_\theta^{-1}(z)$ is the inverse CDF of $\mu^{\mathcal{M}_1}$. Using the assumption that the diameter of $\mu^{\mathcal{M}_1}$ is at most $R$, we have:

$$\mathbb{E}\left[\max_{\theta \in \mathbb{R}^d | \|\theta\|_2 \leq 1} \int_0^1 |F_{m,\theta}^{-1}(z) - F_\theta^{-1}(z)| dz | X_{1:m}, Y_{1:m}\right]$$

$$= \mathbb{E}\left[\max_{\theta \in \mathbb{R}^d | \|\theta\|_2 \leq 1} \int_{-\infty}^\infty |F_{m,\theta}(y) - F_\theta(y)| dy | X_{1:m}, Y_{1:m}\right]$$

$$\leq R\mathbb{E}\left[\sup_{y \in \mathbb{R}, \theta \in \mathbb{R}^d | \|\theta\|_2 \leq 1} |F_{m,\theta}(y) - F_\theta(y)|\right]$$

$$= R\mathbb{E}\left[\sup_{C \in \mathcal{B}} |\hat{\mu}_m^{\mathcal{M}_1}(C) - \mu^{\mathcal{M}_1}(C)|\right],$$

where $\mathcal{B} = \{x \in \mathbb{R}^d | \theta^\top x \leq y\}$ is set of half spaces for $\theta$ and $y$. Since the Vapnik-Chervonenkis (VC) dimension of $\mathcal{B}$ is upper bounded by $d + 1$ (Wainwright, 2019), applying the VC inequality results:

$$\mathbb{E}\left[\sup_{C \in \mathcal{B}} |\hat{\mu}_m^{\mathcal{M}_1}(C) - \mu^{\mathcal{M}_1}(C)|\right] \leq C\sqrt{\frac{(d+1)\log m}{m}},$$

for a constant $C > 0$. Absorbing constants, we obtain the final result. Therefore, we conclude the proof.

## D    IMPLEMENTATION DETAILS

**Network architecture.** First of all, for the white matter segmentation model, we train a vanilla 3D U-Net model with sequential 3D convolutional layers with kernel size $3 \times 3 \times 3$. Secondly, for the deformation network, for each $v_0 \in \mathbb{R}^3$, we linearly transform it to a 128-dimensional feature vector. For each $v_0$, we find a corresponding cube size $5 \times 5 \times 5$ on $\mathbf{I}$. This process can be repeated across multiple scales of $\mathbf{I}$, resulting in the extraction of multiple cubes. Then, we apply a 3D convolution layer followed by a linear layer to transform the spatial cubes to a 128-dimensional feature vector, i.e. same as the feature of $v_0$. Once having the $v_0$'s features and its corresponding spatial features, we concatenate them as a new feature before passing through an MLP, namely $\mathbb{F}_\phi$, to learn the deformation. As discussed, we represent the predicted mesh $\hat{\mathcal{M}}$ and the target mesh $\mathcal{M}^*$ as probability measures $\mu^{\hat{\mathcal{M}}}$ and $\mu^{\mathcal{M}^*}$, respectively. In practice, we can substitute discrete probability measures, e.g. oriented varifold, as $\tilde{\mu}^{\hat{\mathcal{M}}}$ and $\tilde{\mu}^{\mathcal{M}^*}$, respectively. The loss function is computed as follows:

$$\mathcal{L}(\hat{\mathcal{M}}, \mathcal{M}^*) = \widehat{SW}_p^p(\tilde{\mu}^{\hat{\mathcal{M}}}, \tilde{\mu}^{\mathcal{M}^*}) = \frac{1}{L}\sum_{l=1}^L \mathbf{W}_p^p(\theta_l \sharp \tilde{\mu}^{\hat{\mathcal{M}}}, \theta_l \sharp \tilde{\mu}^{\mathcal{M}^*}),$$

where $\mathbf{W}_p^p(\theta_l \sharp \tilde{\mu}^{\hat{\mathcal{M}}}, \theta_l \sharp \tilde{\mu}^{\mathcal{M}^*})$ is the Wasserstein-$p$ (Villani, 2003) distance between $\tilde{\mu}^{\hat{\mathcal{M}}}$ and $\tilde{\mu}^{\mathcal{M}^*}$. We fix $L = 100, p = 2$ for all of our experiments. In terms of oriented varifold, for each support, we concatenate the barycenter of vertices of the face and the unit normal vector as a single vector in $\mathbb{R}^6$. The training procedure is described in Algo. 1.

**Training details.** We optimize both segmentation and deformation networks with Adam optimizer (Kingma & Ba, 2014) with a fixed learning rate $10^{-4}$. We train the segmentation and the deformation networks for 100 and 300 epochs, respectively, and get the best checkpoint on the validation set. All experiments are implemented using Pytorch and executed on a system equipped with an NVIDIA RTX A6000 GPU and an Intel i7-7700K CPU.

## E    DATASET INFORMATION

**Dataset split.** As discussed in Sec. 4.1, we employ three publicly available datasets: the Alzheimer's Disease Neuroimaging Initiative (ADNI) dataset (Jack Jr et al., 2008), the Open Access Series of Imaging Studies (OASIS) dataset (Marcus et al., 2007), and the test-retest (TRT) dataset (Maclaren

---

**Algorithm 1** Training cortical surface reconstruction with SWD distance

---

**Input:** MRI volume $\mathbf{I}$, initial mesh $\mathcal{M}_0 = (\mathcal{V}_0, \mathcal{F})$, learning rate $\eta$, max iter $T$, number projections $L$.

**Initialization:** Deformation network $\mathbb{F}_\phi(\mathbf{I}, \mathcal{M}_0)$.

---

**while** $\phi$ not converge or reach $T$ **do**
  $\nabla_\phi \leftarrow 0$                                             ▷ Zero gradient.
  $\hat{\mathcal{V}} \leftarrow \text{ODESolver}(\mathbb{F}_\phi, \mathcal{V}_0)$                ▷ Deform source vertices $\mathcal{V}_0$ to $\hat{\mathcal{V}}$.
  $\hat{\mathcal{M}} \leftarrow (\hat{\mathcal{V}}, \mathcal{F})$                           ▷ Get predicted mesh.
  $\tilde{\mu}^{\hat{\mathcal{M}}} \leftarrow \text{ToMeasure}(\hat{\mathcal{M}}); \; \tilde{\mu}^{\mathcal{M}^*} \leftarrow \text{ToMeasure}(\mathcal{M}^*)$    ▷ Transform to discrete measures.
  $\nabla_\phi \leftarrow \nabla_\phi + \frac{1}{L}\sum_{l=1}^{L} \nabla_\phi W_p^p(\theta_l \sharp \tilde{\mu}^{\hat{\mathcal{M}}}, \theta_l \sharp \tilde{\mu}^{\mathcal{M}^*})$    ▷ Update gradient.
  $\phi \leftarrow \phi - \eta \cdot \nabla_\phi$                              ▷ Update parameters.
**end while**
**Return:** $\phi_{\mathcal{M} \to \mathcal{M}^*}$

---

et al., 2014). A subset of the ADNI dataset (Jack Jr et al., 2008) is employed, consisting of 419 T1-weighted (T1w) brain MRI from subjects aged from 55 to 90 years old. The dataset is stratified into 299 scans for training ($\approx 70\%$), 40 scans for validation($\approx 10\%$), and 80 scans for testing ($\approx 20\%$). Regarding the OASIS dataset (Marcus et al., 2007), all 416 T1-weighted (T1w) brain MRI images are included. We stratify the dataset into 292 scans for training ($\approx 70\%$), 44 scans for validation ($\approx 10\%$), and 80 scans for testing ($\approx 20\%$). As for the TRT dataset (Maclaren et al., 2014), it consists of 120 scans obtained from three distinct subjects, with each subject undergoing two scans within a span of 20 days.

**Preprocess.** We strictly follow the pre-processing pipeline from (Bongratz et al., 2022). Specifically, we first register the MRIs to the MNI152 scan. After padding the input images to have shape $192 \times 208 \times 192$, we resize them to $128 \times 144 \times 128$. The intensity values are min-max-normalized to the range $[0, 1]$.

## F  VISUALIZATIONS

We provide more visualization of our work as in Fig. 6. We randomly select the prediction meshes from the test set and compute the point-to-surface distance. The color is encoded as how far the point is to the surface. The figures say that the darker color is, the further the predicted mesh to the pseudo-ground truth.

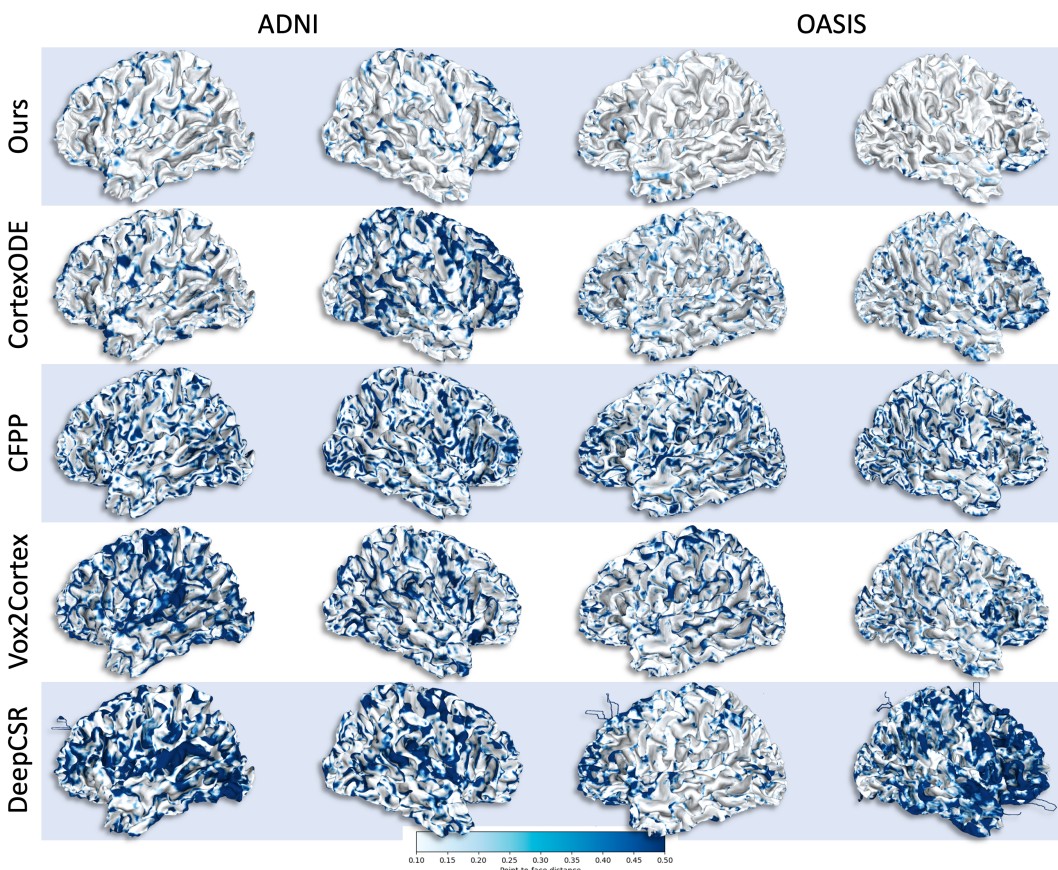

Figure 6: More examples of predicted mesh color-coded with the distance to the ground-truth surfaces as shown in Fig. 2 of the main paper.

