# OpenReview forum: "Diffeomorphic Mesh Deformation via Efficient Optimal Transport for Cortical Surface Reconstruction"
_ICLR.cc/2024/Conference — ICLR 2024 poster_

### Official Review · Reviewer_LCQZ · 2023-10-31

**Soundness:** 3 good
**Presentation:** 2 fair
**Contribution:** 2 fair
**Rating:** 6
**Confidence:** 3

**Summary:**

The paper proposes to learn a diffeomorphic flow using sliced Wasserstein distance (SWD), instead of classical chamfer distance (CD) and earth mover distance (EMD). Such a design choice is not tried before in the context of mesh optimization. The authors shows that SWD leads to better performance, compared to some diffeomorphic flow baselines using other losses.

**Strengths:**

- The proposed solution is simple: just to replace CD with SWD.
- The authors provide theoretical justification for the benefit of using SWD.
- The authors show better quantitative and qualitative results on mesh optimization to reconstruct the cortical structure, compared to some baselines on the same task.

**Weaknesses:**

- The novelty of the idea is limited -- optimizing meshes with CD and/or earth mover distance (EMD) between sampled point clouds is not new (for instance [1]). Replacing EMD with another distributional distance seems more or less some pure trial-and-error endeavor. Indeed, the paper [2] has shown that optimization with SWD on point clouds is beneficial. It is therefore not surprising that SWD can be used for mesh optimization.

- I feel that the math for the probabilistic interpretation, although probably not explicitly presented in the context of mesh optimization, is unnecessary at least from the practical point of view, especially given that EMD has already been used for mesh optimization.

- The presentation of the paper needs to be improved. There are many notations not explained (e.g., the # operator in Eqn. 5).

- While the proposed method seems general, the scope of this paper is quite limited to cortical structures. The authors may consider trying the proposed method on general shapes and compare the results on common benchmarks (if any).

[1] Wang, Weiyue, et al. "3dn: 3d deformation network." Proceedings of the IEEE/CVF Conference on Computer Vision and Pattern Recognition. 2019.

[2] Nguyen, Trung, et al. "Point-set distances for learning representations of 3d point clouds." Proceedings of the IEEE/CVF International Conference on Computer Vision. 2021.

**Questions:**

1. I am not sure if the computation of chamfer distance (CD) should be slower than that of SWD. Some paper argues that the complexity of SWD is similar to CD [2]. And by using KD-tree for nearest neighbor retrieval or some recent methods [3], one should be able to compute CD much faster. I would like the authors explain the speed difference between CD and SWD shown in Fig. 3.

2. There are some improved variants of CD, such as [4]. Just out of curiosity: how do they perform compared to SWD?

[3] Bakshi, Ainesh, et al. "A Near-Linear Time Algorithm for the Chamfer Distance." arXiv preprint arXiv:2307.03043 (2023).

[4] Wu, Tong, et al. "Density-aware chamfer distance as a comprehensive metric for point cloud completion." arXiv preprint arXiv:2111.12702 (2021).

---

> ### Author Response · Authors · 2023-11-17
> **Part1**
>
> We thank Reviewer your time and effort you dedicated to reviewing our paper. We would like to address each of your concerns below.
> > 1. The novelty of the idea is limited -- optimizing meshes with CD and/or earth mover distance (EMD) between sampled point clouds is not new (for instance [1]). Replacing EMD with another distributional distance seems more or less some pure trial-and-error endeavor.
>
> We respectfully disagree with Reviewer in two aspects:
>
> **First of all**,  while acknowledging the prior exploration of optimizing meshes with Chamfer Distance (CD)/Earth Mover's Distance (EMD) between sampled point clouds, we argue that the computation of EMD as an optimization metric is not practical for high-resolution meshes like the cortical surface. We refer the reviewer to the issue: Earth Mover's Distance (EMD) raised on pythorch3D GitHub and its answer by Georgia Gkioxari ([link](https://github.com/facebookresearch/pytorch3d/issues/211)). To the best of our knowledge, the computation of the exact transport map between two point clouds with $n$ points has a polynomial complexity and is not affordable for $n > 20,000$ [5]. The reason [1] successfully employs EMD is due to their experiments on the ShapeNet dataset, which is representable by a few thousand points. In contrast, the complex surface topology of the cortical surface requires mesh approximations with several hundred thousand points.
>
> **Secondly**, we do not replace EMD with another distributional distance due to merely a result of trial-and-error. Specifically, we aim to derive a valid, fast, and scalable metric for high-resolution mesh deformation. While EMD is a valid metric, its applicability to high-resolution meshes is limited, as previously mentioned. On the other hand, Chamfer distance (CD) is faster, but it is not a valid distance since it does not satisfy the identity of indiscernibles, i.e., when CD is 0, two point clouds are not necessarily the same, thus leading to the local minimum issue presented in our paper. Furthermore, as shown in Section 4.2.2 and Figures 3 in our paper, CD (and its variants) is not as scalable as our proposed metric. To the best of our knowledge, we are the first to propose a metric that is valid, fast, and scalable in learning high resolution shape deformation.
>
> > 2. The paper [2] has shown that optimization with SWD on point clouds is beneficial. It is therefore not surprising that SWD can be used for mesh optimization.
>
> While we acknowledge the fact that [2] has shown the optimization with SWD on point cloud is beneficial, it is worth noting the significant differences between point cloud and mesh representations. Loss functions that are effective for point clouds may not directly translate to effective solutions for meshes. Specifically, unlike point cloud presented in [2], the cortical mesh surface we deal with is notably more complex, characterized by its dense connectivity. Moreover, the application of SWD in mesh deformation does not inherently ensure the preservation of the mesh's topological structure. Therefore, our approach, which combines diffeomorphic deformation with SWD loss applied to the probability measures of meshes, is specifically designed to accurately handle complex mesh structures like the cortical surface. This distinction is critical in understanding the novelty and effectiveness of our proposed framework in the context of intricate mesh optimization.
>
> > 3. I feel that the math for the probabilistic interpretation, although probably not explicitly presented in the context of mesh optimization, is unnecessary at least from the practical point of view, especially given that EMD has already been used for mesh optimization.
>
> We respectfully disagree with reviewer for two reasons.
>
> **First**, our probabilistic interpretation, especially our theorem 1, is necessary in the context of this paper. This theorem lays a robust foundation, motivating the representation of meshes as probability measures and justifying the use of Sliced Wasserstein Distance (SWD) as our optimization metric. In fact, from independent point of view, Reviewer **4B9Y** also acknowledges our theorem as "novel and promising" and "suitable for practical applications".
>
> **Secondly**, although EMD has already used for mesh optimization, to the best of our knowledge, there is no theoretical justification for its usage. Furthermore, as presented in Q1, EMD is not a suitable metric for our use case with high-resolution mesh.

---

> > ### Author Response · Authors · 2023-11-17
> > **Part2**
> >
> > > 4. The presentation of the paper needs to be improved. There are many notations not explained (e.g., the # operator in Eqn. 5).
> >
> > The operator $\sharp$ used in Equation 5 has been previously defined in Section 1 under 'Notation.' Allow us to restate the notation here: *We denote $\theta \sharp \mu$ as the push-forward measures of $\mu$ through the function $f:\mathbb{R}^{d} \to \mathbb{R}$ that is $f(x) = \theta^\top x$*. For further notation, we encourage the reviewer to refer back to this section in the main text.
> >
> > We apologize for any confusion regarding notation that may have arisen during the review process. We would greatly appreciate it if the reviewer could point out any notations that are not clearly explained, enabling us to enhance the clarity and comprehensibility of our final draft.
> >
> > > 5. While the proposed method seems general, the scope of this paper is quite limited to cortical structures. The authors may consider trying the proposed method on general shapes and compare the results on common benchmarks (if any).
> >
> > We chose to demonstrate our method using cortical surface reconstruction due to its relevance in neuroscience and neuroimaging. The high resolution of the reconstructed mesh also provides an excellent opportunity to highlight the efficiency of the sliced Wasserstein distance in terms of both scalability and precision. As for its applicability to other datasets, our method is versatile and can be extended to the reconstruction of other organs, including heart [8] and lung [9] reconstruction. Due to the time constrain of discussion process, we are unable to give a comprehensive comparisons on other benchmark. We plan to address this in our future work, expanding the scope of our method's applications and evaluations.
> >
> > > 6. I am not sure if the computation of chamfer distance (CD) should be slower than that of SWD. Some paper argues that the complexity of SWD is similar to CD [2]. And by using KD-tree for nearest neighbor retrieval or some recent methods [3], one should be able to compute CD much faster. I would like the authors explain the speed difference between CD and SWD shown in Fig. 3.
> >
> > In Fig. 3,  we compare the computation of SWD with the implementation of Chamfer distance implemented in Pytorch3D [6], which is employed by all of our competing methods. It is important to note that the Pytorch3D implementation accelerates nearest neighbor (NN) retrieval using custom CUDA kernels, as detailed in Figure 1 of [6]. This implementation takes advantage of parallel computation capabilities on GPUs, a key factor in its efficiency.
> >
> > Regarding KD-tree, the time and complexity of CD using KD-tree is theoretically fast, i.e. $\mathcal{O}(n \log n)$, quite faster than $\mathcal{O}(n^2)$ complexity of the vanilla CD. However, current implementations of KD-trees do not support parallel processing, which limits their scalability, particularly with a large number of points. Additionally, the efficiency of KD-trees diminishes with increasing dimensionality, making them less suitable for GPU computation of high-dimensional data.
> >
> > To validate our claims, we conducted a comparative time analysis of SWD, CD using Pytorch3D, and CD with KD-trees. Since PyTorch doesn't natively support KD-Tree, so we use an external library like [scipy](https://docs.scipy.org/doc/scipy/reference/generated/scipy.spatial.KDTree.html) for this purpose. In low dimensional points, CD-KDtree's speed is comparable to CD-Pytorch3D, but it is still less scalable than SWD. In high dimensional points, e.g. oriented varifold, the speed of CD-KDtree is much slower than both CD-Pytorch3D and SWD. We provide the comparison visualization and our implementation of KD-Tree in the [anonymously link](https://drive.google.com/drive/folders/143lmPXeByrvgsalWxg-r_suOoWagA0Bn).

---

> > > ### Author Response · Authors · 2023-11-17
> > > **Part3**
> > >
> > > > 7. There are some improved variants of CD, such as [4]. Just out of curiosity: how do they perform compared to SWD?
> > >
> > > We thank reviewer for suggesting an interesting work for variant of CD. Density-aware Chamfer Distance (DCD) is proposed in [4] to address the "blindness" of vanilla CD. Currently, we have encountered difficulties in building the CUDA code on DCD codebase on our local machine, which has precluded us from performing a direct comparison. Based on our knowledge derived from the paper, the computational complexity of DCD is similar to that of the standard CD. In fact, Table R5 in [4] indicates that DCD may be even slower than the original CD, suggesting that it is less scalable compared to SWD. That being said, further investigation needed to give a more comprehensive comparison, i.e. in terms of performance and speed, with SWD. We leave it in our future work.
> > >
> > > Should Reviewer have any further queries or concerns, we are more than willing to provide detailed responses to each. Conversely, if our responses have adequately addressed your concerns, we would greatly appreciate it if you could consider revising the score in light of our rebuttal.
> > >
> > > [5] PEYRÉ et al. Computational optimal transport. In Foundations and Trends in Machine Learning, 2019.
> > >
> > > [6] Ravi et al. Accelerating 3D Deep Learning with PyTorch3D.
> > >
> > > [7] Achlioptas et al. Learning representations and generative models for 3D point clouds. In ICML 2018.
> > >
> > > [8] Kong et al. A Deep-Learning Approach For Direct Whole-Heart Mesh Reconstruction.
> > >
> > > [9] Wickramasinghe et al. Voxel2Mesh: 3D Mesh Model Generation from Volumetric Data.

---

> ### Comment · Reviewer_LCQZ · 2023-11-23
>
> I thank the authors' for the extensive response which partially addressed my concerns, mainly those on the empirical results. However, I am still not quite convinced that the theory is that necessary --- from a typical practitioner point of view it does not provide that many additional insights (make a probably not so accurate analogy: one might feel safer to use an algorithm with a better generalization bound but eventually just pick algorithms based on validation error). And what I meant is basically that one may propose the same method by simply noticing the benefits of SWD in point cloud optimization without mathematical verification, and the tone of the paper sounds like the other way. Saying that the authors are inspired by the theoretical framework sounds more comfortable to me. Nevertheless I raise my rating for the empirical contribution of the paper and some previously unexplored theoretical point of view. I would further suggest the authors reduce the amount of math in the main paper and instead elaborate the details in the appendix (for instance, maybe provide an informal version of Theorem 1 in the main paper instead).

---

### Official Review · Reviewer_9F6e · 2023-10-31

**Soundness:** 3 good
**Presentation:** 3 good
**Contribution:** 3 good
**Rating:** 6
**Confidence:** 2

**Summary:**

This paper presents a learning-based diffeomorphic deformation network that employs sliced Wasserstein distance (SWD) as the objective function to deform an initial mesh to an intricate mesh based on volumetric input. Different from previous approaches that use point-clouds for approximating mesh, it represents a mesh as a probability measure that generalizes the common set-based methods. By lying on probability measure space, it can exploit statistical shape analysis theory to approximate mesh as an oriented varifold. It proves a theorem that shows that leveraging sliced Wasserstein distance to optimize probability measures can have a fast statistical rate for approximating
the surfaces of the meshes. The main application is on brain cortical surface reconstruction. Experiment results demonstrate that the proposed method surpasses existing state-of-the-art competing works in terms of geometric accuracy, self-intersection ratio, and consistency.

**Strengths:**

The paper proposes a new metric for learning mesh deformation defined by sliced Wasserstein distance on meshes represented as probability measures that generalize the set-based approach. By leveraging probability measure space, it can gain flexibility in encoding meshes using diverse forms of probability measures, such as continuous, empirical, and discrete measures via varifold representation. The new metric seems novel and works well.

**Weaknesses:**

The paper is very math-heavy and is relatively hard to read for non-experts. The results seem limited to the brain surface. The authors do mention the limitation, which I assume would be a high requirement on the mesh quality. I would like to see more details on that, i.e. how applicable this method is on a common dataset/mesh. The genus zero requirement might be another limitation? can we extend it to shapes of other topology?

**Questions:**

The genus zero requirement might be another limitation? can we extend it to shapes of other topology?

---

> ### Author Response · Authors · 2023-11-17
>
> We thank reviewer your time and effort you dedicated to reviewing our paper. We would like to address each of your concern below.
> > 1. The results seem limited to the brain surface. The authors do mention the limitation, which I assume would be a high requirement on the mesh quality. I would like to see more details on that, i.e. how applicable this method is on a common dataset/mesh.
>
> Our decision to showcase the proposed method through cortical surface reconstruction is motivated by its applicability in neuroscience and neuroimaging fields. Furthermore, the fine resolution of the reconstructed mesh allows us to exhibit the effectiveness of the sliced Wasserstein distance, especially in aspects of scalability and accuracy. Regarding applicability on other dataset, our method can be applied to other organ reconstruction problems such as heart reconstruction [1] or lung reconstruction [2].
>
> > 2. The genus zero requirement might be another limitation? can we extend it to shapes of other topology?
>
> Yes, we agree that, as currently presented in this paper, our method is restricted to genus-zero shapes. The exploration of other shape topologies is a subject for our future research endeavors.
>
> [1] Kong et al. A Deep-Learning Approach For Direct Whole-Heart Mesh Reconstruction.
>
> [2] Wickramasinghe et al. Voxel2Mesh: 3D Mesh Model Generation from Volumetric Data.

---

### Official Review · Reviewer_4B9Y · 2023-11-01

**Soundness:** 3 good
**Presentation:** 3 good
**Contribution:** 3 good
**Rating:** 8
**Confidence:** 4

**Summary:**

This work introduces a learning-based diffeomorphic deformation network that employs sliced Wasserstein distance as the objective function to deform an initial mesh to a complicated mesh based on volumetric input. The work proposes to present triangle meshes as probability measures that generalize the common set-based approach in a learning-based deformation network; the work proposes the sliced Wasserstein distance as a metric  for learning mesh deformation, and proved the convergence rate is solely determined by the number of samples, independent of the dimensionality; the work conducts extensive experiments on white matter reconstruction by employing neural ODE, which show the proposed method outperforms the SOTA in terms of geometric accuracy, self-intersection and consistency.

**Strengths:**

1. This work has solid theoretic foundation, especially the theorem 1 is novel and promising, which shows sliced Wasserstein distance metric   has faster convergence rate than others, and suitable for practical applications.
2. The experimental results are thorough and convincing, the reconstruction mesh quality is good for geometric analysis purposes.
3. The work is well written, the representation is clear, the logic is clean, and the theoretic deduction is explained in details.

**Weaknesses:**

The proposed method to represent a mesh as a distribution in the position-orientation space, and the optimal transportation map is carried out in this space. It should be explained the cost function, and also this representation depends on the position and the orientation of the mesh, and the Wasserstein distance varies if one mesh is transformed by a rigid motion.

**Questions:**

1. What is the cost function defined on the position-orientation space?
2. Does the Wasserstien distance vary when one mesh is transformed under a rigid motion ?
3. Why use the position-orientation space to represent the mesh measure? Why not just use position space?
4. Can we say something about the regularity of the optimal transportation map? Is it diffeomorphic ?

---

> ### Author Response · Authors · 2023-11-17
>
> We thank reviewer your time and effort you dedicated to reviewing our paper. We would like to address each of your concerns below.
>
> > 1. What is the cost function defined on the position-orientation space?
>
> The cost function defined on the position-orientation space is delienated at Appendix D. Specifically, the cost is computed as the Monte Carlo sampling of sliced Wasserstein distance between the predicted mesh $\mathcal{\hat{M}}$ and the target mesh ${\mathcal{M}^\ast}$ represented as probability measures $\mu^{\mathcal{\hat{M}}}$ and $\mu^{\mathcal{M}^\ast}$, respectively: $$\mathcal{L}(\mathcal{\hat{M}}, \mathcal{M}^\ast) = \widehat{SW}\_p^p (\tilde{\mu}^{\mathcal{\hat{M}}},\tilde{\mu}^{\mathcal{M}^\ast}) =  \frac{1}{L}\sum\_{l=1}^L \text{W}\_p^p (\theta\_l \sharp \tilde{\mu}^{\mathcal{\hat{M}}},\theta\_l \sharp \tilde{\mu}^{\mathcal{M}^\ast}),$$
>
> where $\text{W}_p^p (\theta_l \sharp \tilde{\mu}^{\mathcal{\hat{M}}},\theta_l \sharp \tilde{\mu}^{\mathcal{M}^\ast})$ is the Wasserstein-$p$ [1] distance between $\tilde{\mu}^{\mathcal{\hat{M}}}$ and $\tilde{\mu}^{\mathcal{M}^\ast}$, and $\theta\_1,\ldots,\theta\_L$ ($L$ is the number of projections) are drawn i.i.d from $\mathcal{U}(\mathbb{S}^{d-1})$. We empirically set the hyperparameter by tuning on a small subset of OASIS dataset. To be specific, we set $L=100, p=2$. In terms of oriented varifold, for each support, we concatenate the barycenter of vertices of the face and the unit normal vector as a single vector in $\mathbb{R}^6$.
>
> > 2. Does the Wasserstien distance vary when one mesh is transformed under a rigid motion ?
>
> It is possible that the Wasserstein distance changes when one mesh is transformed under rigid motion. Therefore, during training we do not apply any transformation to keep Wasserstein distance stable.
>
> > 3. Why use the position-orientation space to represent the mesh measure? Why not just use position space?
>
> By relying on position-orientation space, we can employ oriented varifold theory, which gives us a theoretical bound [2] for approximating discrete mesh. Furthermore, our experimental results demonstrate that using position-orientation space yields better performance compared to position space only (see Sec. 4.3. Ablation). Therefore, in this paper, we rely on position-orientation space to represent the mesh measure.
>
> > 4. Can we say something about the regularity of the optimal transportation map? Is it diffeomorphic ?
>
> We are assuming that you are asking whether the optimal transport map is a diffeomorphism. It is important to clarify that our approach does not seek to establish an optimal transport map between two manifolds. Instead, we utilize the sliced Wasserstein distance to determine the OT cost. This method primarily serves as a loss function, aiding in quantifying the discrepancy between the predicted and target mesh. Consequently, given the focus of our methodology on sliced Wasserstein distance for loss computation, we cannot definitively ascertain whether the OT map resulting from this process is a diffeomorphism. We are happy to discuss more if Reviewer is still not satisfied with our answer.
>
> [1] Villani et al. Topics in optimal transportation. In American Mathematical Soc., 2003
>
> [2] Kaltenmark et al.  A general framework for curve and surface comparison and registration with oriented varifolds. In CVPR 2017.

---

### Official Review · Reviewer_As4U · 2023-11-01

**Soundness:** 3 good
**Presentation:** 3 good
**Contribution:** 3 good
**Rating:** 8
**Confidence:** 4

**Summary:**

This work addresses the challenge of efficiently measuring the discrepancy between predicted and target 3D meshes, a key component in various 3D vision tasks. The paper introduce a novel metric for learning mesh deformation, defined by the sliced Wasserstein distance. This metric operates on meshes represented as probability measures, which generalize the set-based approach. This approach offers computational efficiency, flexibility in encoding mesh representations, and outperforms other methods in cortical surface reconstruction experiments.

**Strengths:**

This work introduces a novel representation for triangle meshes as probability measures, which generalizes the common set-based approach within a learning-based deformation network. To be precise, the paper outlines three forms of mesh representation as probability measures: continuous, empirical, and discrete measure through the utilization of oriented varifolds. And the authors present a novel learning-based framework DDOT for Diffeomorphic mesh Deformation framework via an efficient Optimal Transport metric, which leverages an efficient Optimal Transport metric. DDOT enables the learning of continuous dynamics to smoothly deform an initial mesh into a complex shape.

The paper is presented in a clear and structured manner, making it easy for readers to understand the proposed methods and experimental results. The work demonstrats improved performance over existing state-of-the-art methods in experiments on multiple brain datasets, particularly in terms of EMD, SWD, ASSD, CN an SI.

**Weaknesses:**

The authors depict a mesh as an oriented varifold, a pivotal concept in this paper. However, for enhanced clarity and accessibility, it's better to provide a concise introduction to oriented varifolds, even though the foundational idea is rooted in earlier works. This brief overview will promote a fundamental understanding of oriented varifolds and their importance in our study, improving the paper's readability.

**Questions:**

1. Is the initial surface in your method the same as the initial surfaces used in competing methods, which are also extracted from the white matter segmentation mask of the brain MRI image?
2. How about the topological information of the initial surface and the ground truth surfaces? Are they all genus-0 surfaces? Does the DDOT framework maintain the topological information during the deformation process?
3. It is better to give a brief introduction of oriented varifold for paper's readability.

---

> ### Author Response · Authors · 2023-11-17
>
> We thank Reviewer your time and effort you dedicated to reviewing our paper. We would like to address each of your concerns below.
> > 1. Is the initial surface in your method the same as the initial surfaces used in competing methods?
>
> In the process of generating initial surfaces from MRI images, each of our competing methods uses a different initial surface. The initial surface in our approach, as well as in CortexODE, is derived from the segmentation of MRI images. Conversely, CFPP and Vox2Cortex commence with a pre-defined mean shape template. DeepCSR, on the other hand, employs an implicit-based methodology and thus does not rely on a template for initiation.
>
> > 2. How about the topological information of the initial surface and the ground truth surfaces? Are they all genus-0 surfaces? Does the DDOT framework maintain the topological information during the deformation process?
>
> Yes, the topological information of initial surface and the groundtruth surface is the same, which is genus-0 surfaces. The utilization of the diffeomorphic deformation framework, DDOT, ensures the preservation of this topological information throughout the deformation process.
>
> > 3. It is better to give a brief introduction of oriented varifold for paper's readability.
>
> We thank Reviewer for the valuable suggestion. In response, we have included a concise overview of oriented varifolds in Appendix A. Additionally, a detailed derivation of mesh as an oriented varifold can be found in Section 3.1. We believe that our current information about oriented varifold sufficiently conveys our concepts and remain easily comprehensible for scholars familiar with the field.

---

> > ### Comment · Reviewer_As4U · 2023-11-23
> >
> > I appreciate the authors' response to my comment. At this time, I don't have any further questions. Thank you.

---

### Author Response · Authors · 2023-11-17

We thank all reviewers for their insightful comments. We appreciate that they find our paper "novel" (**As4U**, **4B9Y**, **9F6e**) and "clear and structured manner" (**As4U**). Furthermore, they also find our proposed method is supported by a "solid theoretic foundation" and "experimental results are thorough and convincing" (**4B9Y**).

Regarding other concerns raised by each reviewer, we will address them below.

---

### Author Response · Authors · 2023-11-21

Dear Reviewers,

We truly appreciate your time and effort in reviewing our paper. Since the deadline is approaching, we really want to hear back from Reviewers if you have any other concerns regarding our rebuttal. We are more than happy to discuss more if Reviewers have any other questions or concerns about our manuscript. Again, we deeply thank the Reviewers for your time and consideration.

Best,

Authors.

---

### Meta-Review · Area_Chair_LTxW · 2023-12-17

**Metareview:**

The paper presents a novel approach (in particular, a novel metric) to learn mesh deformation. Results are shown for cortical surface reconstruction with state-of-the-art performance. Reviewers agree that this work is novel, has a solid theoretical foundation, and claims are supported by empirical results. Furthermore, the authors did a good job of answering comments and addressing concerns during the rebuttal. For those reasons, I do recommend acceptance but encourage the authors to incorporate the reviewer's suggestions on possibly presenting the material in a less dense manner (if possible).

**Justification For Why Not Higher Score:**

Although this is a well-written paper with novelty, the application of cortical surface reconstruction is within a rather small sub-community. As such, this work is possibly of limited interest to a broader community. For spotlight or oral, I would have expected a broader range of applications, demonstrating that the approach is applicable to a wider range of problems.

**Justification For Why Not Lower Score:**

This is a well-written paper with novelty and sufficient experiments; a clear acceptance in my point of view.

---

### Decision · Program_Chairs · 2024-01-16

Accept (poster)